# Do Physical Fitness and Executive Function Mediate the Relationship between Physical Activity and Academic Achievement? An Examination Using Structural Equation Modelling

**DOI:** 10.3390/children9060823

**Published:** 2022-06-02

**Authors:** Adrià Muntaner-Mas, Emiliano Mazzoli, Gavin Abbott, Myrto F. Mavilidi, Aina M. Galmes-Panades

**Affiliations:** 1GICAFE “Physical Activity and Exercise Sciences Research Group”, Faculty of Education, University of Balearic Islands, 07122 Palma, Spain; aina.galmes.panades@gmail.com; 2PROFITH “PROmoting FITness and Health Through Physical Activity” Research Group, Sport and Health University Research Institute (iMUDS), Department of Physical Education and Sports, Faculty of Sport Sciences, University of Granada, 18011 Granada, Spain; 3Institute for Physical Activity and Nutrition, School of Exercise and Nutrition Sciences, Faculty of Health, Deakin University, Geelong, VIC 3220, Australia; e.mazzoli@deakin.edu.au (E.M.); gavin.abbott@deakin.edu.au (G.A.); 4School of Education, University of Wollongong, Wollongong, NSW 2522, Australia; myrto@uow.edu.au; 5Early Start, Faculty of Social Sciences, University of Wollongong, Wollongong, NSW 2522, Australia; 6CIBEROBN “CIBER Consortium of Physiopathology of Obesity and Nutrition”, Carlos III Health Research Institute (ISCIII), 28029 Madrid, Spain

**Keywords:** motor activity, academic performance, fitness, cognition, educational achievement

## Abstract

*Background*: Physical activity health benefits are widely known. However, the association between physical activity, physical fitness, executive function, and academic performance need further investigation. Additionally, one of the literature gaps reveals scarce and mixed findings on what mediators of physical activity may affect academic achievement. *Purpose*: This investigation aims to provide knowledge about the mediation role of physical fitness and executive function in the association of physical activity with academic achievement in a cohort of Spanish schoolchildren using a structural equation modelling approach. *Methods*: The data for this cross-sectional study were collected from a convenience sample of children from Spain (Balearic Islands) aged between 9 and 13 years. Physical activity levels were self-reported with the Physical Activity Questionnaire for Children; physical fitness was assessed using the International Fitness Scale; executive function was assessed with the Trail Making Test, and children’s achievements were collected from the school records. Structural equation modelling was used to explore the relationship between physical activity, physical fitness, executive function, and academic achievement. *Findings*: Statistically significant positive direct associations were observed between physical activity and physical fitness (β = 0.46, 95% CI [0.29, 0.64]), physical fitness and executive function (β = 0.28, 95% CI [0.04, 0.52]), and executive function and academic achievement (β = 0.46, 95% CI [0.28, 0.65]), while adjusting for the confounding effects of sex and body mass index. Furthermore, indirect associations were observed between physical activity and executive function mediated by physical fitness (β = 0.13, bias-corrected 95% CI [0.02, 0.31]) and between physical fitness and academic achievement through executive function (β = 0.13, bias-corrected 95% CI [0.03, 0.32]). *Conclusions*: This investigation adds to the literature with evidence supporting the idea that regular PA leads to improvements in physical fitness and may support cognitive skills and academic performance in children.

## 1. Introduction

Physical activity (PA) refers to any bodily movement produced by skeletal muscles, requiring energy expenditure beyond a basal level. The physical health benefits of PA are widely acknowledged over the lifespan, starting from young children (e.g., improved skeletal and bone health, physical fitness, reduced risk of obesity, and type 2 diabetes) [1]. However, only a very small proportion of children worldwide reach the levels of PA that are recommended for the health-related benefits [2,3]. A bulk of evidence points out PA as a promising and low-cost approach to improving children’s cognitive functions (especially executive functioning) and academic achievement [4,5].

Executive functioning includes a set of cognitive skills that are responsible for goal-oriented behaviour, broadly classified as inhibition, working memory, and cognitive flexibility [6]. The results from systematic reviews and meta-analytic syntheses generally highlighted a positive relationship between PA and executive function. For example, Lubans et al. [7] estimated the effects of PA on executive function to be of small to moderate size in children (effect size [ES] = 0.13–0.57), with acute transient effects of single PA bouts appearing to be greater than chronic, longer-lasting, effects of extended PA practice. Moreover, a direct relationship between executive function, and academic achievement has been investigated [8,9].

In youth, academic achievement is among the most studied constructs associated with cognitive (e.g., executive function) and psychological skills (e.g., goal setting, stress management, self-regulation) and has been considered of the utmost importance for navigating the challenges faced across the child’s lifetime [10,11]. Albeit some systematic reviews and meta-analytic investigations have suggested that PA interventions are associated with enhanced academic achievement, the literature has not yet reached a consensus [12,13,14,15,16].

Physical fitness is a set of attributes that determines a person’s ability to perform PA, e.g., flexibility, motor fitness, cardiorespiratory fitness, or muscular fitness components [17]. PA can improve physical fitness, and greater physical fitness levels have been associated with better executive function and enhanced academic achievement in children [4,11,13,14,15,16,18].

One of the literature gaps reveals scarce and mixed findings on what factors mediate the association between PA and academic achievement. To date, the association between PA and academic achievement in youth has been suggested to be underlain by some mediators, such as physical fitness, adiposity, cognitive function, behaviour, and mental well-being [19]. To our knowledge, only four studies have investigated whether executive function mediated the association between PA and academic achievement in youth. Two cross-sectional studies [20,21] found positive associations, but one longitudinal study and one randomised controlled trial found no significant association [22,23]. There is a need to elucidate how specific executive function domains play a role in this association. In addition, the relation between PA and academic achievement has been analysed in isolation or considering only one mediator at one time; indeed, the interrelation of more than one factor in the same model, such as executive function and physical fitness – to the best of the authors’ knowledge – has not yet been examined. Additionally, only three studies have investigated the mediating effects of physical fitness in the association between PA and academic achievement [21,24,25], two of which reported significant mediating effects [21,24]. Hence, there is a need to study whether the association between PA and academic achievement is mediated by physical fitness components (not only cardiorespiratory fitness) and executive function, using a single model.

The current investigation builds on the previous research/theories wherein PA, executive function, physical fitness, and academic achievement all interact in multiple ways [26,27]. Based on the existing evidence of positive associations between PA and physical fitness [17,28], between physical fitness and executive function [7], and between executive function and academic achievement [8,29], we hypothesise that significant positive associations between PA and academic achievement may be observed through mediating pathways, but not via a direct path. Our study aims to further our understanding of the relationships between the above factors going beyond piecemeal evidence obtained with pairwise associations or simple mediation models. Using structural equation modeling (SEM), we tested, in a cohort of Spanish schoolchildren, a three-path mediated link that integrated all four factors in a serial fashion to provide insight about the mediating role of physical fitness and executive function in the association between PA and academic achievement.

## 2. Methods

### 2.1. Study Design and Recruitment

The data for this cross-sectional study were collected from a convenience sample of children from Spain (Balearic Islands) aged between 9 and 13 years. The participants were recruited during the second term of 2019. All grade four (three classes), five (three classes), and six (two classes) students from one public school were invited to participate. The assessments for each class took place on the same day. The school principal provided informed written consent for the school to be involved in the study. Parents/guardians provided informed written consent on behalf of their children to be part of this study. Children provided verbal assent to confirm their willingness to participate on the assessment day. We received written parental consent for 137 children who completed all the outlined assessments self-reported PA, self-reported physical fitness, and executive function. Seven participants were excluded due to incomplete data for at least three physical education classes. In total, a sample of 130 children was used for this investigation. This study received ethical clearance from the Human Research Ethics Commission of the University of Balearic Islands (reference number: 108CER19) and was conducted in line with the ethical standards outlined by the Declaration of Helsinki.

### 2.2. Measures

The data collection was conducted in a quiet room in the school premises, under the supervision of one member of our research group in the presence of at least one physical education teacher. All of the data on the main variables of interest (i.e., PA, physical fitness, and executive function) used in this study were collected on the same day, except for academic achievement. Participants’ academic achievement was obtained from the school records of the Balearic Island School Register during the second school term.

The data on the potential confounders were measured on the assessment day (i.e., child height and body weight) or collected via a demographic survey that parents/guardians of participating children were invited to complete at the time of consent. Parental surveys included questions on the child’s age, sex, and parental education. Other information (e.g., child grade, classroom, and academic achievement) was collected via the school.

### 2.3. Self-Reported PA

PA levels were self-reported with the PA Questionnaire for Children (PAQ-C) validated for the Spanish population [30,31]. The PAQ-C is a 10-item, 7-day PA recall; however, the last item is not used in the calculation of the PA score. In the first PAQ-C item, children were asked to recall their participation in several activities (e.g., bicycling, jogging/running, dance) over the last 7 days, with responses provided on a scale from 1 (not performed) to 5 (undertaken seven times or more). Items 2–4 presented questions relating to school-based PA (i.e., physical education and activity during recess/lunch) with possible responses ranging from 1 (I don’t do physical education/I sat down) to 5 (Always/Ran and played most of the time). Items 5–7 relate to participation in sports and PA outside school hours (i.e., right after school, evenings, and weekends) with possible responses between 1 (none) to 5 (6 times or more). Item 8 asked children to decide which of the five statements reflected their activity in the last 7 days (e.g., “All or most of my free time was spent doing things that involved little physical effort” [1] to “I very often (7 times or more) did physical things in my free time” [5]). Item 9 asked children to recall how much activity they performed for each day of the week, with possible responses between 1 (none) and 5 (very often). Item 10 asked the child to indicate whether they were sick in the last week or could not participate in PA for other reasons. The mean of the first nine items was calculated to produce a PAQ-C activity summary score.

### 2.4. Self-Reported Physical Fitness

Children’s perception of their physical fitness was assessed using the International Fitness Scale (IFIS) [32]. The instrument has been validated for Spanish children [33]. The questionnaire asks participants to reflect and provide a rating on their physical fitness for each of the following five domains: (i) the overall fitness, i.e., ‘my general physical fitness is…’; (ii) cardiorespiratory fitness, i.e., ‘my cardiorespiratory fitness (capacity to do exercise, for instance, long-running) is…’; (iii) muscular fitness, i.e., ‘my muscular strength is…’; (iv) speed-agility, i.e., ‘my speed/agility is…’; and (v) flexibility—i.e., ‘my flexibility is…’. Each item was scored on a five-point Likert scale (ranging from 1, “very poor”, to 5, “very good”).

### 2.5. Executive Function

The executive function were assessed with the Trail Making Test (TMT), a pen and paper test often used to measure inhibition–attention and cognitive flexibility [34,35] but also often referred to as a central measure of executive function, as its completion involves inhibition, working memory and cognitive flexibility. The TMT comprises five different conditions: (1) visual scanning, (2) number sequencing, (3) letter sequencing, (4) number-letter switching, and (5) motor speed [36]. The first three conditions and the fifth one assess visual-perceptual abilities, whereas condition four is an indicator of cognitive flexibility. The times taken to complete each condition were measured. Visual scanning was used to test the participants’ ability to find objects and requires participants to visually scan letters and numbers on the provided sheet and quickly mark only the numbers. The number sequencing condition requires participants to draw a line to connect numbers 1–25 in ascending order as soon as possible and was used to assess visual–perceptual skills. In the letter sequencing, participants should connect presented letters in alphabetical order as quickly as possible; this part was used to evaluate basic sequencing. Number-letter switching requires drawing a line alternating between numbers and letters and following an ascending/alphabetical order (e.g., 1-A-2-B) as quickly as possible. The last condition requires participants to quickly draw a line over a dotted line that connects the same number of objects presented in the other conditions and was used to evaluate visual and motor speed. For the analysis, the recorded time at the number sequencing condition was subtracted from that of the number–letter switching condition and used as a measure of executive functioning.

### 2.6. Academic Achievement

Children’s achievements in Math, Spanish Language, Catalan Language, Grade Point Average (GPA), and Physical Education (PE) were collected (on a scale from 0 [worst] to 10 [best]) from the school records at the second term as assessed by teachers.

### 2.7. Potential Cofounders

Potential confounders included age (in years), sex, body mass index (BMI), and parental education. Information on children’s age and sex were collected upon consent via their parents. Children’s body weight and height were collected with an electronic scale (TANITA BC 601 Ltd., Paris, France) and a stadiometer (SECA 213 Ltd., Hamburg, Germany), respectively. For these assessments, the children were required to wear lightweight clothes and no shoes; measurements were taken twice and recorded to the nearest 0.1 kg/cm. Children’s weight and height were used to calculate BMI (i.e., BMI = kg/m^2^).

Parental education was reported through questionnaires completed by children’s parents/guardians. For the analysis, information on parental education was categorised as follows: (i) no university, (ii) university level (one parent/guardian), and (iii) university level (both parents/guardians).

### 2.8. Statistical Analyses

The descriptive statistics of all the variables were calculated for all of the children and split by sex. The data were assessed for normality, and transformations were considered for variables that did not satisfy the normal distribution assumptions. Structural equation modelling was used to explore the relationship between PA, physical fitness, executive function, and academic achievement. The hypothesised model included the relationships presented in Figure 1A. Latent factors ‘physical fitness’ and ‘academic achievement’ were each represented by five indicator variables in the hypothesised model, while ‘PA’ and ‘executive function’ were single-indicator latent factors. To account for potential confounders, the basic model (Figure 1A) was integrated with the relationships depicted in the hypothetical model in Figure 1B.

For the latent variables ‘physical fitness’ and ‘academic achievement’, items showing too poor of a loading (i.e., λ < 0.40) were removed from the final model. For parsimony, potential confounding variables that showed little evidence (*p*-value > 0.2) of a relationship with the main constructs of interest (i.e., physical fitness, executive function, and academic achievement) were removed from the final model. As the latent variables ‘PA and ‘executive function’ each included only a single indicator variable, error variances for these indicator variables were set to be equal to [(1−reliability)×variance] [37], using the sample variances and test-retest reliability previously reported for each of the measurements (*r* = 0.72 for the PAC-Q score and *r* = 0.80 for TMT). The model fit was determined based on the comparative fit index (CFI) and the Tucker–Lewis index (TLI) (CFI and TLI values ≥ 0.9 are indicative of acceptable model fit), and the Root Mean Square Error of Approximation (RMSEA; good model fitted indicated by values < 0.06) [38]. Since a few (*n* = 8) children had missing data for one variable but complete data for the remaining variables, the maximum likelihood with missing values (MLMV) estimation method was preferred over the traditional maximum likelihood (ML) approach, to utilise all the available data. Correlations between indicator variable error terms were allowed when modification indicated possible improvements in the model fit and made theoretical sense (e.g., cardiovascular endurance and strength within fitness). Indirect associations were examined to understand whether: (i) physical fitness mediated the association between PA and executive function; (ii) executive function mediated the associations between physical fitness and academic achievement; and (iii) physical fitness and executive function mediated the association between PA and academic achievement. For each estimated indirect effect, we used bootstrapping (1000 resamples) to calculate bias-corrected 95% confidence intervals (CI). All of the analyses were conducted using Stata Statistical Software (StataCorp, 2021. *Stata Statistical Software: Release 17*. College Station, TX: StataCorp LLC).

## 3. Results

The 130 children who completed the assessments and were included in the main analysis of this investigation were aged 10.69 ± 0.96 years (57% boys). The children’s characteristics are summarised in Table 1.

All of the data were approximately normally distributed. Since the loadings for flexibility (Physical Fitness → IFIS 5) and physical education (Academic achievement) were too low (i.e., both λ < 0.20), these variables were removed from the final model. The GPA variable was highly correlated (r > 0.84) with the other remaining Academic performance variables, suggesting that it did not uniquely contribute to the factor. This makes logical sense considering that this variable is a grand average of all school subjects. For this reason, only Math, Spanish Language, and Catalan Language were maintained as indicator variables for academic achievement. Age and parental education were removed from the model as they did not show evidence of relationships with any of the variables of interest (*p*-values for age = 0.51–0.99; *p*-values for parental education = 0.53–0.83). The final model is presented in Figure 2.

The model showed statistically significant positive direct associations between PA and physical fitness (β = 0.46, 95% CI [0.29, 0.64]), physical fitness and executive function (β = 0.28, 95% CI [0.04, 0.52]), and executive function and academic achievement (β = 0.46, 95% CI [0.28, 0.65]), while adjusting for the confounding effects of sex and BMI. No other statistically significant direct associations were found. Significant indirect associations were observed between PA and executive function mediated by physical fitness (β = 0.13, bias-corrected 95% CI [0.02, 0.31]) and between physical fitness and academic achievement through executive function (β = 0.13, bias-corrected 95% CI [0.03, 0.32]). Despite the aforementioned statistically significant direct and indirect associations observed, there was not a significant indirect association between PA to academic achievement via physical fitness and executive function (β = 0.01, bias-corrected 95% CI [−0.18, 0.17]). The standardised direct and indirect structural coefficients for the final model are presented in Table 2.

Model fit statistics comparing the final model to some alternative model options, including those fitted using maximum-likelihood where eight observations were excluded due to missing data, are presented in Table 3. Notably, the observed structural coefficients and significance levels remained relatively stable regardless of the different models examined (results not shown), and the fit statistics indicated an acceptable to a good fit for all of the models.

## 4. Discussion

This study aimed to verify the hypothesis that children who are more physically active may also perform better academically and test whether such improvements are direct or mediated through children’s physical fitness and/or executive function. To the best of the authors’ knowledge, this is the first study that attempts to verify this empirically using SEM. The overall results of our study did not confirm the hypothesised association chain from PA to academic achievement through physical fitness and executive function after controlling for confounders. However, other important results were found: (i) a positive direct association between PA and physical fitness; (ii) a positive direct association between physical fitness and executive function; (iii) a positive direct association between executive function and academic achievement; (iv) a positive indirect association between PA and executive function mediated by physical fitness; (v) a positive indirect association between physical fitness and academic achievement through executive function.

Our results did not provide evidence of a direct association between PA and academic achievement, which is in line with some findings from previous studies testing the impact of PA on children’s school performance [25,39,40,41,42,43]. In contrast, there is a large body of evidence stemming from systematic reviews endorsing positive acute and chronic effects (small to moderate effect sizes) of PA participation on academic achievement in children [12,13,14,15,16]. In our investigation, a self-reported PA measure was used and partly explained this disagreement with the bulk of the literature. Moreover, a recent systematic review and realist synthesis demonstrated that several factors at individual, task and context level may act as moderators of the relationship between PA and cognitive outcomes [44]. This complexifies the pattern of relations and may explain inconsistent results of primary studies and reviews. Additionally, our results are in line with previous research reporting a direct and positive association between PA and physical fitness in children [14,45]. By contrast, our data did not confirm a link between physical fitness and academic achievement, which is opposite to the current literature [18,46,47].

One of the key questions in the literature is whether the association between PA and academic achievement is mediated by physical fitness. Our results did not confirm this idea but showed an indirect association between PA and executive function mediated by physical fitness. Compared to available scientific evidence, some cross-sectional studies on children and adolescents have reported mediated associations of PA with academic achievement via cardiorespiratory fitness [21,41,42,48,49] and others did not [22,25,39]. The longitudinal study of Ishihara et al. [50] found that sports participation was positive but indirectly related to academic achievement 2-year later via cardiorespiratory fitness in 463 12–13-years-old children. Additionally, to the best of the author’s knowledge, there exists only one randomised controlled trial in the field, and they found that cardiorespiratory fitness mediates the effects in only one arm of the PA intervention on academic achievement in students from 30 Norwegian lower-secondary schools [24].

Some hypotheses could explain our findings in context with the evidence available. The available research in the field has utilised a heterogeneity of methodologies to ascertain and direct association of physical fitness with academic achievement and the mediating role of fitness in the PA-academic achievement relationship. Specifically, the methods of previous studies have varied from the use of direct/reported PA or physical fitness measures, the use of direct or indirect measures, the inclusion of samples with children and adolescents with different age ranges, the analysis of only one physical fitness component (cardiorespiratory fitness), the conduction of the research in an ecological context (school-based) or laboratory conditions.

Our findings did not show a direct relationship between PA and executive function, which is not in line with current evidence supporting the view that both acute and chronic PA is effective for improving executive function in children [7]. However, our data showed an indirect association between PA and executive function via physical fitness, suggesting that PA might indeed affect executive function only to the extent to which it improves physical fitness. It is worth mentioning that the majority of the studies in the field have used inhibition tasks, and far less frequently cognitive flexibility tasks. Thus, inconsistencies between the outcomes of the present and previous studies may be an issue of specificity of effects for the different subdomains of executive function [51]. However, our study demonstrated a positive and direct association between executive function and physical fitness, which is accordant with the literature [11,52]. Likewise, a direct and positive association was encountered between executive function and academic achievement in our study (β = 0.46). This result is similar to the findings of Cortés Pascual, Moyano Muñoz, and Quílez Robres [29], who encountered a small effect size (*r* = 0.37) in their meta-analyses. The findings from our study also align with the largest meta-analyses in the field in which 299 studies and 65,605 elementary school-age children were analysed, and a direct and positive relation was found between executive function and academic achievement [8].

Lastly, our findings do not support a hypothesised model in which executive function mediates the relations between PA and academic achievement. However, the current study adds value to the existing literature by analysing the indirect association included in the same model of two different mediators not previously explored, the physical fitness and executive function. To date, and to the best of our knowledge, only a few studies have assessed the mediating effects of executive functions with mixed results [20,21,22,23,53]. Amongst those, the only PA intervention study included 1129 10-year-old children and followed them for over 7 months. The authors concluded that the intervention effects on academic achievement was not explained by a change in executive function [23]. The considerable differences in the methodological approaches used across these studies (e.g., the use of direct/indirect measures of PA, different measures of executive functioning, and varying indicators academic achievement measured with different instruments) hamper us from establishing direct comparisons;. Our failure to identify a mediation role of executive function might also be explained by the use of different statistical approaches to examine these relations (i.e., fewer studies have utilised SEM). Collectively, with the past knowledge and our findings, it seems uncertain the extent to which PA impacts academic achievement via executive function, given that the results have varied from trivial null effects to small positive effects.

There is available evidence from systematic reviews and meta-analyses investigating the chronic effects of PA on academic achievement [54,55] and executive function [5,56] in children and adolescents. However, the exact nature of this relationship and its possible mediators are still unclear [57]. The current study attempted to explore these questions using SEM. Thus, the novelty and innovative design are considered the strengths of this study.

Nevertheless, also limitations need to be acknowledged. Firstly, this study was cross-sectional and based on a convenience sample of modest size. Future research should include longitudinal and experimental studies, especially randomised controlled trials, for more robust results. Importantly, the current study used self-reported measures for PA and physical fitness. Whilst we know that such measures tend to be prone to bias, we accounted for measurement error in our SEM model. Directly measured PA through accelerometers [58] and physical fitness tests [59] may provide more accurate representations of these constructs. Similarly, the current study utilised proxy-reported measures of academic achievement, which warrant predictive validity. Future research can use more valid and reliable tools such as standardised assessments (e.g., the Program for International Student Assessment [PISA]). Furthermore, the vast majority of research assessed cognition with tests of core executive functions (i.e., working memory, inhibition, and cognitive flexibility) [11,60]. So far, only one review included neuroimaging studies to evaluate the effects of acute PA on cognition across the lifespan and not specifically targeting children [61]. Including other measures of cognitive function might also be insightful. Currently, studies exploring the effects of PA on brain structure or function have been conducted outside the school environment (e.g., lab or afterschool programs) [61], except for one study testing the effects of school-based active breaks on behavioural and brain outcomes [62]. Different contexts may trigger or not trigger different mediators acting on the relationship of PA to cognition and academic achievement [44].

## 5. Conclusions

Overall, the present study adds to the literature with evidence supporting the idea that regular PA is associated with improvements in physical fitness and may support cognitive skills and academic performance in children. Despite our study failing to support the hypothesis of a direct/indirect relationship between PA and academic achievement, we found direct associations between PA and physical fitness, physical fitness and executive function, and executive function and academic achievement. In addition, indirect associations emerged from PA to executive function through physical fitness and from physical fitness to academic achievement through executive function. Future interventional studies should verify whether PA interventions aimed at improving physical fitness and/or executive function through a combination of high-intensity and cognitively engaging PA can also lead to improvements in academic achievement. As an intervention strategy, PA is low-cost, has little or no side effects, and has the potential to benefit many children, both physically (strong evidence) and cognitively (trivial to small positive evidence). It would be appropriate to develop and test forms of activities that can be directly delivered at and by schools in addition to the curricular physical education, which would be an equitable approach to reach a larger audience.

## Figures and Tables

**Figure 1 children-09-00823-f001:**
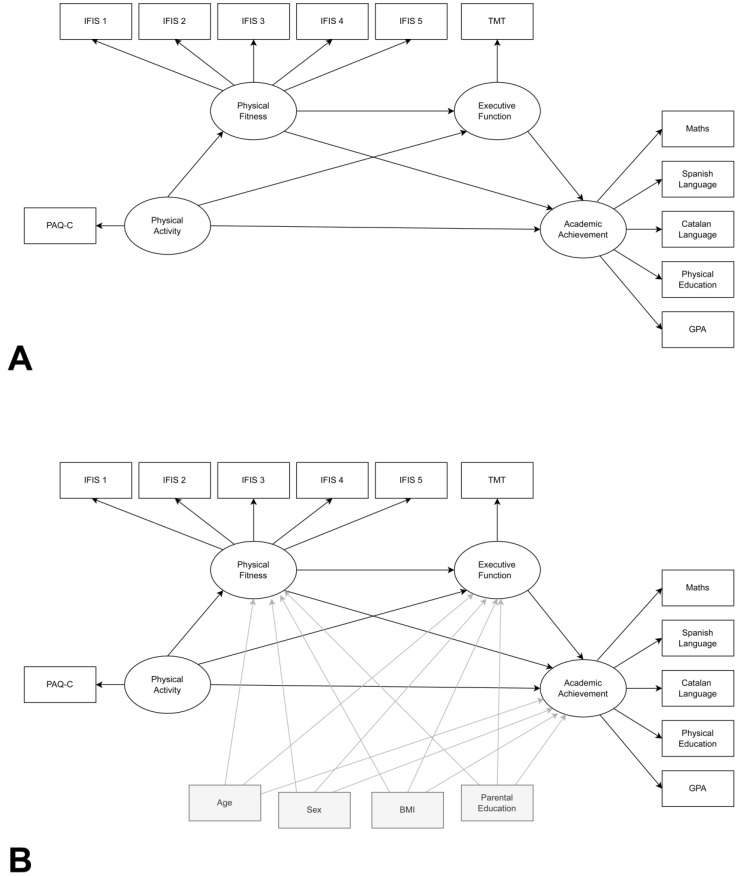
Hypothesised model of the relationships between physical activity, fitness, executive function and academic achievement without (**A**) and with confounding variables (**B**)**.** Physical Activity Questionnaire for Children (PAQ-C); International Fitness Scale (IFIS); Trial Making Test (TMT); Body Mass Index (BMI). IFIS 1. Overall fitness; IFIS 2. Cardiorespiratory fitness; IFIS 3. Muscular fitness; IFIS 4. Speed-agility; IFIS 5. Flexibility. Grade Point Average (GPA).

**Figure 2 children-09-00823-f002:**
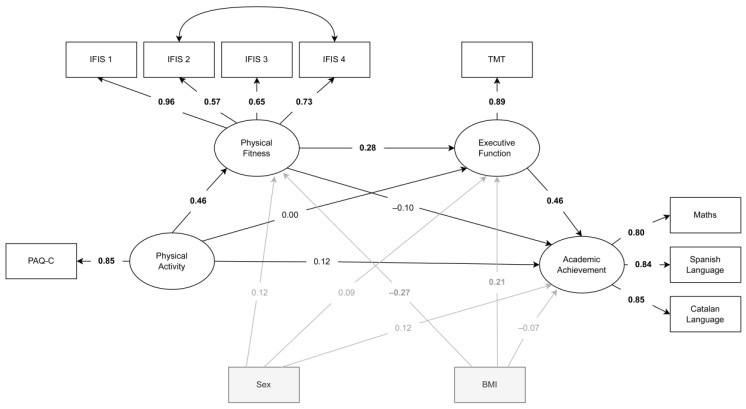
Final model. Physical Activity Questionnaire for Children (PAQ-C); International Fitness Scale (IFIS); Trial Making Test (TMT). Note that the TMT score was reversed so that a greater value reflected better performance. Body Mass Index (BMI). IFIS 1. Overall fitness; IFIS 2. Cardiorespiratory fitness; IFIS 3. Muscular fitness; IFIS 4. Speed-agility. Significant associations are marked in bold.

**Table 1 children-09-00823-t001:** Children’s demographic characteristics for the entire sample and divided by sex (*n* = 130).

	All		Girls		Boys		Sex Comparison	
Characteristics	*n*	%/M ± SD	*n*	%/M ± SD	*n*	%/M ± SD	Statistic	*p*-Value
*Sex*								
Female	56	43						
Male	74	57						
Age (years)	130	10.69 ± 0.96	56	10.70 ± 0.95	74	10.69 ± 0.98	t = 0.04	0.967
*School grade*							χ^2^ = 0.84	0.657
Grade 4	51	39	24	43	27	36		
Grade 5	40	31	15	27	25	34		
Grade 6	39	30	17	30	22	30		
*Parental education*							χ^2^ = 0.36	0.834
No university	56	43	23	41	33	45		
University level (one parent)	45	35	21	38	24	32		
University level (both parents)	29	22	12	21	17	23		
Body Mass Index (kg/m^2^)	130	17.90 ± 2.85	56	17.66 ± 2.88	74	18.08 ± 2.83	t = −0.81	0.417
Self-reported physical activity (score range 1–5)	122	2.80 ± 0.57	53	2.79 ± 0.61	69	2.82 ± 0.55	t = −0.30	0.768
*Self-reported physical fitness (score range 1–5)*								
IFIS 1. Overall fitness	130	3.90 ± 0.80	56	3.80 ± 0.82	74	3.97 ± 0.78	t = −1.20	0.235
IFIS 2. Cardiorespiratory fitness	130	3.77 ± 0.87	56	3.68 ± 0.83	74	3.84 ± 0.89	t = −1.05	0.297
IFIS 3. Muscular fitness	130	3.81 ± 0.87	56	3.77 ± 0.95	74	3.84 ± 0.81	t = −0.44	0.660
IFIS 4. Speed–agility	130	3.89 ± 0.82	56	3.82 ± 0.81	74	3.95 ± 0.83	t = −0.86	0.392
IFIS 5. Flexibility	130	3.28 ± 1.03	56	3.21 ± 1.09	74	3.34 ± 0.98	t = −0.67	0.506
*Executive function*								
Trail Making Test (s)								
1. Visual scanning	130	69.08 ± 7.69	56	69.29 ± 9.07	74	68.93 ± 6.51	t = 0.25	0.805
2. Number sequencing	130	60.11 ± 8.63	56	61.48 ± 9.27	74	59.07 ± 8.01	t = 1.56	0.122
3. Letter sequencing	130	68.90 ± 6.42	56	69.95 ± 7.01	74	68.11 ± 5.85	t = 1.59	0.115
4. Number-letter switching	130	140.58 ± 39.91	56	147.05 ± 45.40	74	135.68 ± 34.72	t = 1.56	0.122
5. Motor speed	130	63.02 ± 9.14	56	64.66 ± 9.59	74	61.77 ± 8.64	t = 1.77	0.079
Interference score (i.e., 4 minus 2)	130	80.47 ± 36.52	56	85.57 ± 40.27	74	76.61 ± 33.16	t = 1.35	0.179
*Academic performance (score range 0–10)*								
Maths	130	6.54 ± 1.41	56	6.27 ± 1.48	74	6.74 ± 1.32	t = –1.92	0.057
Spanish Language	130	7.04 ± 1.16	56	6.86 ± 1.17	74	7.18 ± 1.15	t = 1.42	0.167
Catalan Language	130	7.17 ± 1.24	56	7.00 ± 1.33	74	7.30 ± 1.16	t = 1.43	0.167
PE	130	8.12 ± 1.03	56	8.13 ± 0.97	74	8.12 ± 1.07	t = 1.44	0.167
GPA	130	7.22 ± 0.92	56	7.06 ± 0.95	74	7.33 ± 0.89	t = 1.45	0.167

Note: mean (M); standard deviation (SD); International Fitness Scale (IFIS).; grade point average (GPA). physical education (PE).

**Table 2 children-09-00823-t002:** Standardised direct and indirect association coefficients of the final model.

Variables	β	*p*-Value	95% CI	
Direct Associations				
*Physical fitness*				
Physical activity	**0.46**	**<0.001**	**0.29**	**0.64**
*Executive function* ^a^				
Physical activity	0.00	0.991	–0.26	0.27
Physical fitness	**0.28**	**0.020**	**0.04**	**0.52**
*Academic achievement*				
Physical activity	0.12	0.350	–0.13	0.37
Physical fitness	–0.10	0.395	–0.34	0.13
Executive function	**0.46**	**<0.001**	**0.28**	**0.65**
Indirect associations ^b^				
Physical activity → Fitness → Executive function	**0.13**	-	**0.03**	**0.32**
Fitness → Executive function → Academic achievement	**0.13**	-	**0.02**	**0.31**
Physical activity → Fitness → Executive function → Academic achievement	0.01	-	–0.18	0.17

Statistically significant (*p* < 0.05) associations are marked in bold. ^a^ Note that the Trail Making Test score—used as an indicator of executive functioning—was reversed so that a greater value reflected better performance. ^b^ 95% confidence intervals for indirect associations were calculated with asymmetric, bias-corrected bootstrapping (1000 resamples), hence *p*-values were not reported. Confidence intervals (CI).

**Table 3 children-09-00823-t003:** Goodness of fit statistics for structural equation models testing the hypothesised association between physical activity, physical fitness, executive functions, and academic achievement (*n* = 130).

Model	Method	Obs.	χ^2^ (*p*-Value)	RMSEA [90% CI]	AIC	BIC	CFI	TLI	CD
Final Model	MLMV	130	47.01 (0.042)	0.060 [0.012, 0.095]	4531.90	4660.94	0.968	0.946	0.785
	ML	122	46.65 (0.046)	0.062 [0.009, 0.098]	4260.20	4372.36	0.967	0.944	0.792
Model A	MLMV	130	52.89 (0.068)	0.053 [0.000, 0.086]	4454.47	4563.43	0.970	0.957	0.740
	ML	122	50.83 (0.097)	0.050 [0.000, 0.085]	4204.03	4310.58	0.972	0.961	0.740
Model B	MLMV	130	92.34 (0.022)	0.054 [0.022, 0.080]	5973.38	6168.37	0.947	0.922	0.790
	ML	122	93.01 (0.019)	0.057 [0.024, 0.083]	5606.99	5758.41	0.943	0.915	0.797

Maximum Likelihood with Missing Values (MLMV); Maximum Likelihood; Root Mean Squared Error of Approximation (RMSEA); confidence interval (CI); Akaike’s information criterion (AIC); Bayesian information criterion (BIC); Comparative fit index (CFI); Tucker-Lewis index (TLI); coefficient of determination (CD). The preferred model is Final Model using MLMV, which was obtained excluding poor factor loading and including only covariates (sex, BMI) for which there was evidence at the *p* < 0.2 level of relationship with the main constructs of interest (i.e., physical fitness, executive function, or academic achievement). Model A reflects the hypothesised model without GPA as a factor loading of academic achievement and without potential confounders. Model B reflect the hypothesised model without GPA as a factor loading of academic achievement with all potential confounders included. All models allowed for correlation between indicator variable error terms, where modification indices indicated significantly improved model fit and deemed theoretically appropriate and were conducted using two estimation methods (MLMV or ML).

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
