# Peer review of "Do Physical Fitness and Executive Function Mediate the Relationship between Physical Activity and Academic Achievement? An Examination Using Structural Equation Modelling"

_children, 2022, doi:10.3390/children9060823_

Round 1

Reviewer 1 Report

Thank you for the opportunity to review this manuscript. It is about the study of correlation between physical activity and academic achievement and if this correlation is direct or mediated by other factors. Even if the topic is not innovative, a structural equation modelling has been adopted making the study interesting. The manuscript is well written and presented, the methodology is well structured and the statistical analysis is complete and satisfactory. The introduction is clear and complete as the discussion. After minor corrections I strongly suggest the publication of this manuscript.

Line 27: Please, provide what SEM means.

Line 29-31: please avoid the use of abbreviations in the abstract

Line 37: Please, provide what BMI means.

Line 78: please, adopt always the abbreviations. Physical activity: PA. Please, double check the manuscript.

Considering the model adopted, I strongly suggest to consider the study: Tabacchi et al. An Interaction Path of Mothers’ and Preschoolers’ Food- and Physical Activity-Related Aspects in Disadvantaged Sicilian Urban Areas . Int. J. Environ. Res. Public Health 2021, 18, 2875. https://doi.org/10.3390/ ijerph18062875

Author Response

Dear Reviewer,

Thank you very much for your revisions. Minor revisions have been considered and the text has been amended. 

Sincerely,

Adrià Muntaner-Mas on behalf of co-authors

Reviewer 2 Report

Review of the article Do physical fitness and executive functions mediate the relationship between physical activity and academic achievement? An examination using Structural Equation Modeling

Abstract

Written succinctly and to the point.

Introduction

The introduction is based on the latest scientific reports, which emphasizes the high nature of the work. Correct justification of the purpose of the work. Correct discussion and use of the integrated model for analyzes.

Materials and Methods

Well-chosen research tools.

The authors write ... "Data for this cross-sectional study was collected from a convenient sample of children 106 from Spain (Balearic Islands) aged between 9 and 13 years" ... - please justify choosing such a group?

Well-chosen statistical models.

Results

Own results well presented and described.

Discussion

There were interesting but rather obvious discoveries in the discussion that PA has a close relationship with physical fitness. But it is physical fitness that derives from physical activity that has positive associations with various executive functions.

Moreover, very careful analysis and comparison to other studies is very much in the discussion.

Conclusions

A rich literature completes this good article

Author Response

Dear Reviewer,

Thank you very much for your revision report.

The decision to include children aged between 9 and 13 years was driven for one specific reason. The age group selection was made on the basis that during late childhood the brain undergoes development and thus is a sensitive period of maturation. Further, we selected a child sample instead of an adolescent because physiological and psychological changes are dramatic, and the speed of changes/maturation differs a lot among individuals, and it is difficult to control these processes and confounding factors.

Sincerely,

Adrià Muntaner-Mas on behalf of co-authors